# Female community health volunteers' experience in navigating social context while providing basic diabetes services in western Nepal: Social capital and beyond from systems thinking

Usha Dahal[1☺¤], Rekha Lama Tamang[1☺*], Tania Aase Dræbel[1], Dinesh Neupane[2,3], Sweta Koirala Adhikari[2], Pabitra Babu Soti[2], Bishal Gyawali[1]

**1** Global Health Section, Department of Public Health, University of Copenhagen, Copenhagen, Denmark,
**2** Nepal Development Society, Kaski, Nepal, **3** Department of International Health, Johns Hopkins Bloomberg School of Public Health, Johns Hopkins University, Baltimore, MD, United States of America

☺ These authors contributed equally to this work.
¤ Current address: Institute of Family Medicine and Public Health, Institute of Social Studies, University of Tartu, Tartu, Estonia
* rekhaltamang@gmail.com

## Abstract

The global burden of non-communicable diseases (NCDs) has led to an increased mobilization of community health workers (CHWs) in the prevention and management of NCDs, particularly in resource-poor settings. However, little is known about the experiences of CHWs as they navigate the complex social context while proving home-based NCD management. This study aims to explore the experiences of female community health volunteers (FCHVs) in a community-based pilot project in western Nepal, specially regarding the social challenges they face while delivering basic type 2 diabetes (T2D) services. Using a qualitative phenomenological approach, the study conducted two focus group discussions and nine in-depth interviews with a total of 14 and 9 FCHVs, respectively. Social Capital theory was employed to understand the sociological aspects. The findings shed light on the challenges encountered by FCHVs in expanding their social networks, building trust, and fostering reciprocity among T2D intervention recepients. Notably, social trust was a significant challenge, compounded by power dynamics related to gender and socioeconomic status. FCHVs managed to overcome these challenges through their perseverance, self-motivation, and leaveraging their strong bonding and linking social capital. The recognition they received from the community played a crucial role in sustaining their motivation. The study highlights the importance of FCHVs' strong social capital, supported by available resources and personal motivation, in overcoming social obstacles. It is imperative for community health interventions to anticipate challenges across various elements of social capital to ensure the long-term retention and motivation of CHWs. Establishing appropriate support systems that address personal motivating factors and the strengthen social capital is essential.

**Data Availability Statement:** All relevant data are available from within the manuscript.

**Funding:** The study was conducted for the purpose of a Master's thesis by UD and RLT based on one of the projects of World Diabetes Foundation (WDF), Denmark entitled 'Diabetes prevention and management by lay health workers in Nepal'- project number WDF16-1441 implemented by Nepal Development Society- NeDS. The thesis was submitted to the Global Health Section, Department of Public Health at the University of Copenhagen. Data collection was funded by WDF, grant number WDFI16-1345_1-2021. The funders did not interfere with the study design, tools and techniques, interpretation of data, manuscript preparation and publication (https://www.worlddiabetesfoundation.org/).

**Competing interests:** The authors have declared that no competing interests exists.

## Introduction

Non-communicable diseases (NCDs) have become one of the major causes of death and disease burden causing 71% of global deaths of the 57 million deaths which occurred globally in 2016 [1]. The NCD burden is greatest in low-and middle-income countries (LMICs), where 78% of all NCD deaths and 85% of premature deaths occur [1]. Among NCDs, diabetes mellitus stands as a prominent segment, and in 2019, it was estimated to cause approximately 1.5 million deaths worldwide [2]. Notably, type 2 diabetes (T2D) accounts for around 90% of all diabetes cases. The rise in diabetes burden has challenged the healthcare systems and policymakers in LMICs to explore innovative solutions to effectively address the threat posed by this disease [3]. Community health workers (CHWs) have played a significant role in combating communicable diseases and providing maternal and child health services, especially in resource-constrained settings [4, 5]. CHWs act as a bridge between the community, healthcare systems, patients, and practice teams to manage chronic diseases as suggested by Chronic Care Model (CCM) [6]. The model has proven to be an effective tool to improve the practice of chronic illness management in healthcare systems as it identifies the challenges and opportunities of involving CHWs at three levels: health system, community, and CHWs themselves [6]. Evidence from LMICs shows that CHWs are now considered valuable human resources for combating NCDs [3, 6, 7], including type 2 diabetes (T2D) [8–10]. Moreover, in some settings, CHWs are also treated as a part of a multidisciplinary intervention team of doctors, nurses and other paramedics who provide specialist health care and support [8]. However, there is no consensus on defined roles of CHWs in community-based health programs [11], especially in resource-poor settings [3].

CHWs' roles and activities are reported to vary in the worldwide context [11, 12]. While performing their roles, CHWs encounter several support systems and challenges. The most-reported support mechanisms for CHWs at a community level in the various studies included community embeddedness, personal development, altruism, social prestige, and financial incentives [5, 11, 13–17]. On the other hand, the challenges most frequently mentioned in these studies include the lack of resources to meet the communities' health and non-health needs and employees' expectations; difficulties in providing health services; low financial incentives; inadequate programmatic, logistic, training, and supervision support; long travel time; insufficient recognition for their work; and the struggle to balance their own livelihood and community service [14, 18–21]. Besides, despite the individual autonomy that CHWs possess, challenges remain in sustaining their motivation due to the voluntary or low-paid nature of their work and ensuring the quality and accuracy of their task [6]. In spite of the progress made in involving CHWs in addressing NCDs, particulary T2D, there remains a dearth of research on the social challenges they encounter in real-life situations. Limited literature explores how CHWs position themselves within the intricate social context while dealing with these challenges, highlighting a noticeable research gap [14, 18]. To bridge this gap, we formulated a research question: What kind of social complexities do CHWs face, and how do they navigate these complexities while fulfilling their role in the community to prevent type 2 diabetes?

Our study is based on the community-based pilot project, entitled 'Diabetes prevention and management by lay health workers in Nepal', which aimed to raise awareness of T2D among community people, identify undiagnosed people with prediabetes and T2D, train female community health volunteers (FCHVs) on home-based blood sugar screening, and provide health education [22].

Nepal is no exception to the rising global burden of diabetes [2], as Gyawali et al. showed the burden of T2D ranging from 1.4%-19% among adults with a pooled prevalence of 8.4%,

and can be effectively managed through the invovlment of FCHVs at the community level [9, 23]. FCHVs are lay health volunteers mobilized by the national health system of Nepal, primarily as a first contact point of health services that bridges community people with the health system to uplift the maternal and child health status, including family planning in their local communities [9, 24]. They are selected by the consensus of Mothers' Group (MG) which is a committee formed among the local women in each ward to upgrade the health status of community people [4, 25] along with to empower women in the society [26]. When FCHVs are mobilized, they receive minimum daily allowance and transportation costs, but they offer their services on a voluntary basis [4].

Despite the increasing global trend in mobilizing CHWs to combat diabetes, mobilizing FCHVs is in the infancy stage in Nepal [9]. Nevertheless, the results from few interventions revealed the positive outcomes of trained FCHVs' capability in regard to home-based diabetes screening, early detection, case management and lifestyle changes via providing health education [1, 27, 28]. However, at the national level, the roles of FCHVs on T2D and other NCDs management are still unclear, although the Nepal government endorsed the WHO Package of Essential NCD (PEN) package [29] and WHO Heart Technical package [30]. Moreover, in resource-constrainsd countries like Nepal, the efforts to expand NCDs activities are gradual, but there remains a lack of awareness regarding the challenges faced by FCHVs while engaging in community-based NCDs activities, especially in the social factors.

This study aims to explore the experiences of FCHVs in a community-based pilot project in western Nepal, specially regarding the social challenges they face while delivering basic T2D services. This study holds significant practical implications for countries with limited resources like Nepal, providing valuable evidence for policymakers, program managers, and community leaders to design and implement community-based NCD programs, involving CHWs. Moreover, the study aims to address dilemmas related to the roles of FCHVs in NCD prevention and management, especially concerning T2D.

## Methodology

### Study design

A phenomenological qualitative research design was employed in this study to document the lived experiences [31] of FCHVs regarding their involvement in T2D intervention. A phenomenology study was chosen to deeply comprehend the lived experiences and gain a profound understanding of the phenomenon [32]. To ensure objectivity and impartiality, a process of 'Bracketing' was incorporated, where we intentionally put aside our own beliefs, emotions, and opinions to remain open and faithful to the phenomenon being studied [31, 33].

### Study area

The study was carried out in Pokhara Metropolitan City of Kaski district. where T2D intervention was executed through the funding support of the Worold Diabetes Foundation (WDF), and implemented by a local non-government organization named Nepal Development Society (NeDS). To ensure comprehensive coverage, all eight administrative wards (wards from 26 to 33) withian the project's catchment area were purposely selected for the study.

In the T2D intervention program, a total of 168 FCHVs took part in the 5-day training, but only 66 passed the written and practical tests. At the end of the T2D intervention, out of 66 eligible FCHVs only 63 FCHVs performed 10179 home-based diabetes screening and counseling. Among which, 5487 adults (aged 25–64 years) who were found at high risk of diabetes were followed-up for 3 times throughout the intervention period. By the end of March 2022, more than 62,858 people within the intervention area, were sensitized, and made aware about T2D

complications and its management. FCHVs' roles in the T2D intervention were to visit service recipients' households three times a year to provide blood glucose screenings and health education. In addition, they were required to screen blood pressure, measure height and weight, refer patients to nearby health facilities, monitor medication adherence, and inform people about modifiable risk factors for T2D [22].

For the screening of T2D it was important for FCHVs to visit service recipients' homes early in the morning before they break their fasting state to avoid T2D false diagnosis and ensure consistent results [34]. FCHVs were provided with basic equipment such as glucometers, glucometer strips (referred to as sugar kits by FCHVs in this study), blood pressure devices, and height and weight scales [22]. FCHVs were provided with Nepalese Rupees (NPR) 100, i.e., United States Dollar (USD) 0.84 [35] per home visit. In this study, we attempt to address this gap by drawing FCHVs' first-hand experiences regarding their community work in the context of T2D management.

## Study participants and recruitment

Among 66 trained and enrolled FCHVs in the T2D intervention, 23 participants were purposely selected in this study based on inclusion criteria. These criteria included those who had experienced the T2D intervention phenomena, agreed to participate in a lengthy interview, granted consent to record the interview, and were available when the study was conducted [31, 36]. Data were collected in two forms; 2 focus group discussions (FGDs) and 9 in-depth interviews (IDIs), to yield comprehensive stories of their engagement in the phenomenon. FGDs were conducted among 14 FCHVs. Despite setting the estimated sample size in advance, the number of interviews was finalized with respect to the point of data saturation [36].

## Study tools and techniques

Open-ended questionnaires were developed in the forms of IDI and FGD guides to collect data after the rigorous literature review and consultation within the research team guided by CCM model [6]. The interview guides were designed to explore the challenges FCHVs faced and the way they navigated those challenges throughout the T2D intervention emphasizing their experiences, motivating factors, and the working environment. Clear and concrete terms were used in the development of the interview guides [36]. The key words used in the guides were defined and discussed among the research team, including field researchers. Theory-laden questions were avoided in the development of interview guides for both IDI and FGD, as suggested by Bevan (2014) [33]. Data were triangulated from responses obtained from FGDs and IDIs to achieve data richness and an in depth understanding of the phenomena at individual and group levels, enhancing the trustworthiness of study findings [37]. The data collection tools were primarily developed in the English language, which later were translated into the Nepalese version (see S1 and S2 Texts). The interview guides were pre-tested among the non-participants of this study and were revised before the start of the data collection. The interview guides were slightly modified based on the participants' response while interviews were being conducted [36]. The phenomenological reduction (also known as 'Epoche') was adopted during the entire interview process to maintain a fundamental level of validity [33]. Context-specific and imaginative variation questions were used to get closer to the real-life experiences of the participants, improving the credibility of the interviews, in cases where the phenomenon varied within the participants [33].

## Data generation

Two local field researchers with a background in health sciences and experience in qualitative interviewing were trained to conduct data collection. As they collected data, they were

instructed to note any non-verbal communication, including silence, sighs, laughter, facial expressions, and gestures of study participants. The process began with two FGDs, observed by first authors (UD and RLT) via the Zoom platform, who later interacted with the participants to extract further data. This was followed by nine IDIs. Data collection took place from 15 April to 15 May 2021. However, after the COVID-19 lockdown in Nepal, the main authors conducted two online IDIs, and field researchers conducted five follow-up interviews via telephone. The IDIs lasted from 40 to 60 minutes and FGDs from 120 to 160 minutes. In addition, the relevant information gathered from the webpage of the WDF, the meeting minutes, the research progress logbook, the field researchers' field notes, and the gray literature provided by the program implementers (donor and host organization) were used as the sources of information.

## Data analysis

The data analysis in our study began with the first FGD, and continued in a sequential manner [38]. As the study progressed, we had the opportunity to improve and gather high quality data from subsequent FGD and interviews. All IDIs and FGDs were audio-recorded, transcribed, and translated into English. Subsequently, the research team read through the transcripts to become familiar with the data. Inductive content analysis was undertaken to describe and systematically organize the phenomena [39]. We interpreted both the verbal and non-verbal responses to give the meanings of the words during the latent content analysis [39, 40]. Thereafter, we performed open coding using Nvivo software to identify patterns and arranged varied codes systematically [31]. Subsequently, sub-categories were freely generated and grouped by merging the identical ones under several categories to reduce the frequencies [39]. We eliminated the overlapped, repeated, unclear and irrelevant sub-categories to manage data [36]. Thereafter, we combined and thematized the categories with similar meanings from FGDs and IDIs. As we explored sub-themes, our findings gradually formed a cohesive picture, revealing that the challenges faced by FCHVs in T2D intervention were primarily rooted in social phenomena, with power dynamics outweighing technical challenges. Consequently, we recognised the importance of delving deeper into sociological aspects to provide additional explanations. Following a comprehensive review of several sociological theories, we decided to employ Robert Putnam's and other scholars' concepts of the social capital theory, which are widely recognised within the health sciences for understanding health related outcomes globally, to interpret, structure, and analyse the study's findings [41–43]. While Putnam's conceptualization of social network, social trust, and norms of reciprocity closely aligned with our findings, we acknowledged the limitations highlighted by other scholars such as Fine, Szreter and Woolcock, Song et al., and John [42–45]. To address these limitations, we supplemented Putnam's concept by incorporating the contributions of other scholars, primarily Coleman, Bourdieu and Szreter and Woolcock [44, 46, 47].

Social capital theory helped us to interpret the findings on the challenges closely associated with trust, social networks, norms and values [43] that FCHVs encountered while providing T2D services. We used the definitions of bonding, bridging, and linking social capital proposed by Szreter and Woolcock to explain the functioning of FCHVs' social networks, social trust, and norms of reciprocity in a more nuanced manner. Coleman's operationalization of resources facilitated our understanding of the actions of individuals and community members [46], while Bourdieu's conceptualization of power dynamics added a crucial perspective to the analysis of social capital [47]. Furthermore, the concept of systems thinking [48] helped us visualize and analyze the feedback loops among interconnected elements of social capital, including other dominant factors discovered during our analysis. We developed the systems diagrams using *Vensim* [49] and *Kumu* [50] softwares.

## Ethical considerations

The ethical approval for the study was obtained from the ethical review board of the Nepal Health Research Council (registration no. 96/2021 MT). The study was conducted complying the principles of the Declaration of Helsinki [51]. The study participants were informed about the study aims and procedures where their participation was strictly voluntary. Informed written and verbal (in the case of a phone interview) consent was obtained from the study participants before conducting the interviews. Study risks and benefits were explained to the participants before their participation. In addition, the participants were assured that their identities would be kept anonymous throughout the study.

## Results

The study revealed that the elements of social capital, including 1) expanding Social Network; 2) gaining and maintaining Social Trust; and 3) generating Norms of Reciprocity among community people were the major challenges that FCHVs had to face in T2D intervention. In addition, power dynamics challenged FCHVs interplaying with social capital elements (Fig 1).

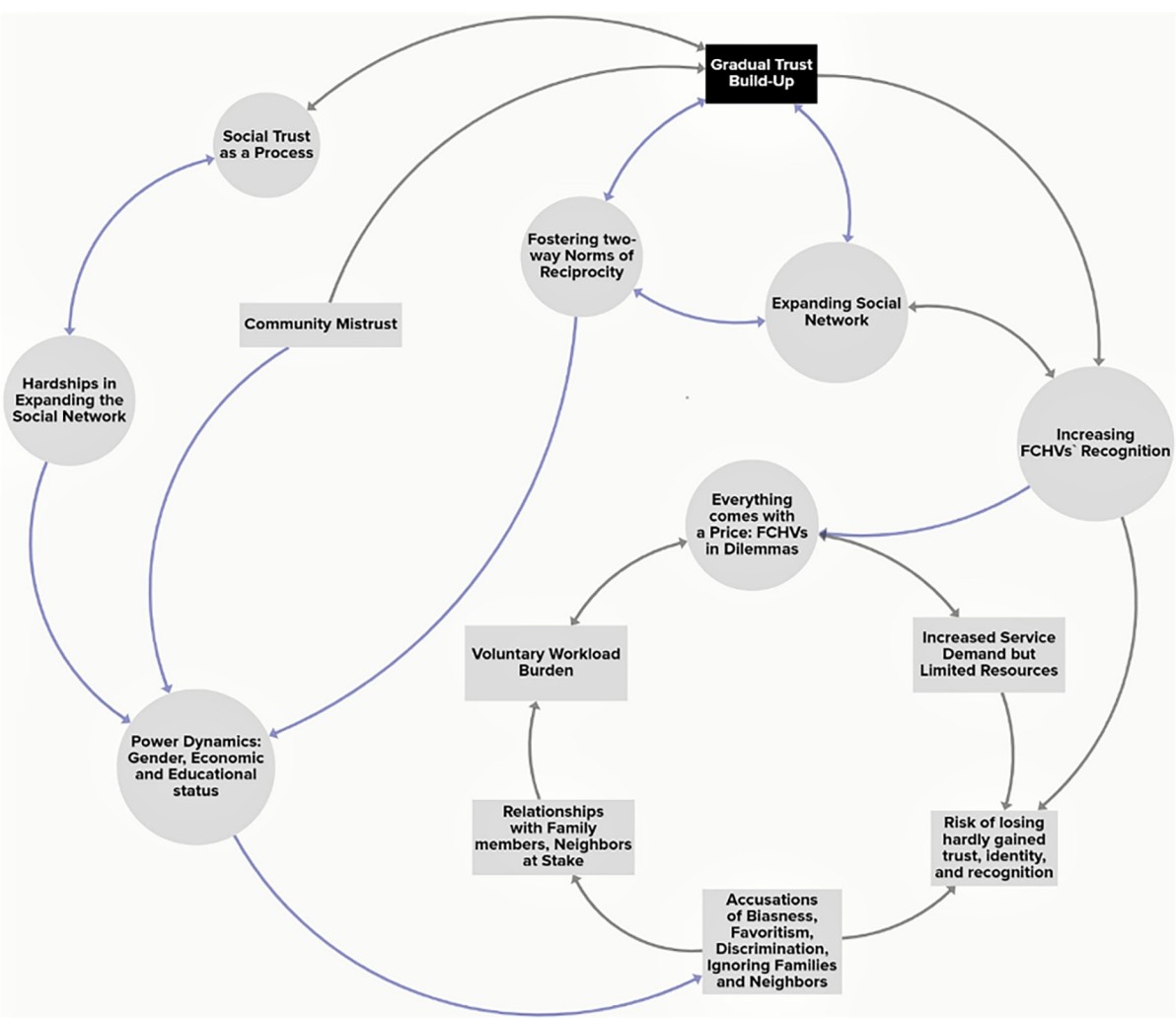

**Fig 1. Dynamics of social capital and its implications on FCHVs' life and work.**

### a. Hardships in expanding the social network

FCHV's primary social network was mostly among women of reproductive age because of their engagement in maternal and child health services under Nepal Government's FCHV program. Since T2D intervention covered a diverse population in terms of age, gender, and geography, expanding the social networks was the first step FCHVs had to accomplish. While expanding their social networks and building relationships with service recipients, trust issues exacerbated by gender-based power dynamics became a major challenge. While sharing their unpleasant experiences, all FCHVs consistently used the "he" pronoun with few exceptions of "she" and "they" indicating men being more skeptical towards FCHVs' skills and being reluctant to accept them as basic diabetes service providers. For example, in FGD-I, one of the participants mentioned:

"**FGD-I, P6**: *He yelled, "Do not come to my home carrying that bag! (FCHVs' tool-kit bag)". So, I did not visit that home thereafter!".*

### b. Social trust as a process

**Mistrust and accusations.**   Community people doubted the very core purpose of FCHVs' voluntary acts, accusing them of working for financial gain. Community people's mistrust was more intense in the beginning compared to the later phase of the program. For instance, one FGD participant explained:

*FGD-II, P4: Some people's perception is totally different. They ask, "how much salary are you receiving?". We go there to provide services, but they think that we don't do home visits or provide services without money, which is why they show us such behaviors.*

Community people mistrusted FCHVs' qualifications and potential to perform the T2D-related activities. However, our study participants had at least a secondary level of education, received pre-service training, and were enrolled in the T2D intervention only after passing the training test. A participant expressed that one of the male community people strongly underestimated her potential, mistrusted the equipment, and accused her of working for the greed of money:

*FGD-I, P8: He (one of the male community people) used to say, "Her salary is secured, why does she need to worry? Uh! This diabetes testing equipment is not good! Why do people have to spend lots of money to study Doctor of Medicine (MD) if we could easily measure (blood sugar levels) here with her (referring to FCHV)?"*

According to FCHVs, male community members not only hindered the FCHVs' performance but also barred their wives from receiving diabetes-related services from FCHVs. One participant in FGD-I shared an experience where she found a woman was prevented to take services from FCHVs by her husband:

*FGD-I, P8: He used to say, "I don't even go to Pokhara (city) for the check-up. Do you think that I will come to you for the check-up?". However, his wife told me she will come for screening if I don't inform her husband. She checked her sugar level with me a few times but later said, "My husband found out that I came to you for the screening; therefore, he scolded me!".*

Such trust issues led people to spread negativity about FCHVs in the community; thus, some people did not welcome them in their houses, and some avoided the services. One discussant summarized such a situation in these terms:

*FGD-I, P6: He (one of the community people) used to spread negativity by saying, "What is the use of her (FCHV)? Again today, she came with her bag (provided by Nepal Government to carry basic health medicines, information education and communication materials)! This equipment (test kits) does not work; I have checked my sugar and pressure with that!". After listening to him, other service recipients who came for the tests used to leave the venue without testing.*

FCHVs' non-verbal signs during the interviews reflected that such accusations were serious, unacceptable, and emotionally draining. Some FCHVs also felt intimidated by community peoples' negative behavior, such as scolding made them mentally and emotionally exhausted. For example, one of the IDI respondents shared how a renowned local leader's harsh words and loud voice traumatized her:

*IDI 1: "My heart was pierced a bit! I felt some trauma, I felt upset. . . It was like that. . . (Becomes emotional). . . as he was shouting too loud."*

**Gradual trust build-up.**   In the later phase of the intervention, community people gradually trusted FCHVs, and started seeking medical advice other than diabetes-related complications. One participant expressed how community people started relying on their health advice:

*FGD-II, P8: Now, they approach us with full trust and share, "I have this (health problem); what should I do now?". Because of trust, they seek advice from us whenever they have other health problems.*

In addition, visits to medical facilities by service recipients after screening with FCHVs generated some level of trust among them, helping to spread the information within their network.

Relevant quote from FGD-I showing gradual trust development towards FCHVs is highlighted below:

*P3: In the start, when I checked the sugar levels, one guy had 12 mmol/l (sugar level), and I advised him, "Your sugar levels are high. You should go to Sishuwa hospital (district-level community hospital)." But he replied egotistically, "No! I will control myself. I will not go anywhere!" And soon after 10 to 15 days, he suffered from high sugar levels and got admitted to hospital. . . After his stay at the hospital for five days, he is very polite whenever we meet (sense of realization)!*

Besides, FCHVs'gradual increase in their competency and confidence over time and strong continuous support from their family and NGO also assisted in gaining the trust of the community people. One FCHVs shared her experience in IDI:

*IDI 8: Before, we had the feeling that we may not be able to perform the tests, they (community people) may scold us and may not cooperate with us. But now, after performing the tests several times, it has become easier. . . It was a bit difficult the first time but became easier the second and third time.*

Such a gradual change in trust was indicated by the community people's willingness to engage and pay for FCHVs' services. A few FCHVs in FGD-I shared that some people raised

their voices in a few MG meetings to have a mass screening and were ready to pay for the necessary costs.

*Conversation among participants in FGD-I:*

**FGD-I, P1:** *The ones who know about it say that we need to check their sugar and pressure also during the MG's meeting.*

**P2:** *They said, "even if it is paid check-ups!". Few people have asked about MG meeting venues so they could do the diabetes screening and pay for it.*

### c. Gradual trust helped in expanding the social network

FCHVs persistently worked to build bridging social capital to expand their social network besides their existing network. In addition, people spread information about FCHVs' work by word of mouth in various community gatherings, which helped amplify trust in FCHVs. One of the FGD-I participants exemplified how gradual trust build-up contributed to the expansion of FCHVs' social network:

**FGD-I, P7:** *When we did the test, some people's tests showed high sugar and pressure levels. And in meetings and gatherings like this, community people share their opinions among them by saying, "Today, she checked my sugar, and it was high. She checks nicely." This is how the discussions start, and we get recognized even more.*

### d. Trust towards FCHVs and their expanded social network increased their recognition

The majority of the FCHVs shared their experiences on how they felt recognized after being engaged in T2D intervention as it helped break the notion carried by the community people about FCHVs having limited skills. The accomplishment of their newly assigned roles fostered community's trust towards FCHVs inducing the expansion of the FCHVs' social network, which according to FCHVs, ultimately benefited them to gain respect and identity. One participant shared her story regarding gaining respect:

**FGD-I, P8:** *Currently, because of the impact of this program, those people who used to ignore us by not even bothering to make an eye contact, after they check their sugar and pressure with us, they are the ones who call and greet us in the first place with respect. This proud moment is what we want. (Smiles). . . At least they realized and showed respect for us.*

### e. Everything comes with a price: FCHVs in dilemmas

Gradual trust development and increased recognition in the community brought other challenges. Demand for T2D screening was heightened, putting FCHVs in dilemmas as they were challenged by the limited resources and time available to meet those demands. Their dilemmas took place in various forms ranging from the risk of losing hardly gained trust and recognition; the accusation of biasness, favoritism, and discrimination, risking their relationships with their family and neighbors; being forced to rely on their own resources; provoking them to question their worthy of volunteerism due to added workload.

**Risk of losing hardly gained trust, identity, and recognition.** FCHVs shared their unpleasant experiences on how increased demands of T2D services from the community people with no mechanism to fulfill those demands risked their hardly gained trust, and recognition in the community. As frontline community health workers, they had to constantly battle to maintain trust and recognition irrespective of the health resources they were provided with by the diabetes intervention. One of the FCHVs reflected the scenario where she was in a difficult position due to increased demand for the T2D services:

> *IDI 6: Although only the husband's name is on the program's list, their wives also come together. They (wives) ask, "Check mine (sugar) as well! Why don't you check?". It is a problem to quarrel with them, but at the same time, it is also a huge problem for me.*

**The new types of accusations raised: biasness, favoritism, and discrimination.** Misunderstanding among both the service providers and community people on the selection process of the service recipients in the diabetes intervention took another turn. On the one hand, FCHVs succeeded in gaining trust and recognition regarding their diabetes services. On the other hand, the majority of FCHVs were accused of being biased and selective in terms of the caste system hierarchy and including only their favorite people in the diabetes intervention. One participant in FGD-II shared:

> *FGD-II, P8:. . .Two of them were arguing with me by saying, "Why did you exclude us? Is it because we are Dalit (lower caste in Caste System)? Why didn't you check ours?". And we had to convince them by saying that their chance will come very soon, and we are just working based on the name list the NGO sent.*

**Relationships with family members and neighbors at stake.** Similarly, the unavailability of extra glucometer strips (these are small plastic strips that help to test and measure blood glucose levels) also induced problems in FCHVs' own families and neighborhoods as their family members and neighbors often felt excluded. In addition, our observation also revealed that FGDs' participants were unhappy and confused while expressing issues of inability to screen their own and family members' blood glucose levels. One of the FGD participants sadly expressed:

> *FGD-II, P3:. . .even during mealtime, my mother-in-law starts scolding me, "What can you do for our family if you do not even have one sugar kit to measure for your own family?" I had to face such an extremely difficult situation when I had to keep those kits at home to test others than my family members.*

**Invested own resources.** To balance the chaotic situation raised by the increased service demand, FCHVs started investing their own resources. They tried their best to save their relationships with the family, neighbors, and community people. For instance, they bought glucometer kits, invested extra time to measure blood pressure, and to convince community people about their next turn.

> *IDI 6: I would not be at peace if I told them that I will not measure theirs as they were my neighbors. If I checked the sugar of 4 people then, I always used to check the blood pressure of an additional 6 to 7 people (to sustain in the community).*

**Burdened by the voluntary work.** FCHVs also felt occupied and burdened from the voluntary work as they had to compromise their personal lives. Few FCHVs also expressed

increased workload, making it further difficult to manage the household chores and perform FCHVs' other regular roles assigned by the Nepal government.

*FGD-I, P5: Everything is fine, and we are very happy, but the ones who come to check their blood pressure when we are in a hurry sometimes irritate us. It becomes difficult for us to decide whether to check or not. If we do not check their pressure, they are our relatives. If they frequently come, especially when we are in a hurry to leave our house, we must stay. It seems like we must stay on standby for the whole day just for this job!*

## f. Balancing norms of reciprocity

According to FCHVs, the efforts contributed by the community people did not match the efforts of FCHVs. Generating norms of reciprocity among the community was a huge challenge to run the intervention smoothly, as diabetes intervention demands both the parties (the service providers and service recipients) to be equally accountable for the significant outcomes of the intervention.

**Power dynamics: Wage-based worker Vs. health prioritization.**   Fasting blood glucose test requires a fasting state, and this method itself was pressuring FCHVs to reach service recipients' homes early in the morning or before they break their fasting state. However, FCHVs faced extreme difficulty in reaching service recipients because either they would leave their homes early in the morning or forget to be in a fasting state. Such a lack in generating feelings of reciprocity among service recipients made it difficult to execute their tasks, forcing them to drop such service recipients from the program. Service recipients who were wage-based workers could not prioritize their health needs.

*FGD-II, P4: Sometimes, we had to go through steep hills, rain, and hot scorching sun. Although we wake up early at 3–4 AM and go to the people's homes, we do not find them because they are wage-based workers. After trying 4 or 5 times, I said, "we are trying really hard to come to your home, but why can't you give us one or half an hour?" Their responses are like, "We must work to live. What are you going to give us (monetary benefits)?" We had to face these extremities.*

**Service recipients kept forgetting their roles.**   Although FCHVs constantly reminded service recipients to stay in a fasting state on the screening day until they arrived, service recipients kept forgetting their roles, complicating the service delivery of T2D. One of the FCHVs clarified this scenario in the interview:

*IDI 8: It takes us 3 or 4 attempts of home visits just for a single test. We must call them a day before a test, and they agree. But when we reach there in the morning, they say, "I already had a tea. Let's do it tomorrow." Then we have to say "okay" and return. It is very hard to work!*

**Expanded social networks and trust fostered reciprocal feelings among service recipients.**   Expansion of FCHVs' social network and gradual trust development facilitated mutual reciprocity among service recipients. This feeling was indicated by the community's gradual support, including service recipients, in several aspects. Some of the service recipients started convincing those who were initially ignorant, arranging a venue to provide services, gathering service recipients at one place for the ease of FCHVs, assisting FCHVs to record the information, and bringing the absent service recipients to the venue.

> *IDI 2: At first, I went to everyone's house. And for the next time, they told me, "Instead of you going to everyone's house, call us a day before we meet so that the next day, we will gather at one place for you."*

### g. Supporting resources to navigate the complex social context

FCHVs had several support systems to navigate such a complex social context. Their motivation was generated based on their feeling of social responsibility, being the most prominent. Likewise, a sense of happiness gained from having an opportunity to save people's lives along with continuous support from their families and the NGO were the crucial resources that FCHVs relied on to navigate the challenging social context.

**Feeling of social responsibility.**    FCHVs felt they should take the social responsibility to serve the community as responsible members of society. An IDI respondent shared how she feels being in her position:

> *IDI 9: One should not only think about family matters and financial incentives when living in a community, especially when one is already committed to help others.*

**Sense of happiness by providing the services.**    The most frequent pattern in this study that FCHVs felt was the right opportunity for them to serve the community and the ultimate satisfaction they sensed, especially when they identified risk groups of diabetes and referred them to the right place for timely treatment.

> *FGD-II, P5: I feel very satisfied when I can provide service to all. I feel like "Sewa nai Dharma ho!" (Literal translation: Service is Religion).*

**Family support.**    Another strong support system was continuous family support, as expressed by all FCHVs. Mostly, their husbands were supportive by assisting them in recording forms and accompanying them in their long-distance travel.

> *FGD-II, P4: Regarding the help from my family, it is my husband who carries my bag. I convinced him I could not go alone because it was far and asked him to come with me. Then we went together.*

**NGO's support.**    Enormous and instant programmatic support from the NGO became a strong support system to navigate the challenges. However, many participants also felt the need for refresher training, coordinating with health workers, and conducting mass awareness from the NGO.

> *IDI 6: NGO has helped me whenever I called them asking about anything; they came to me anytime I needed. That also made it extremely easier to work.*

## Discussion

Our study explored how FCHVs navigated the social context while performing their assigned roles in T2D intervention in western Nepal. The analysis showed that FCHVs' experiences were affected by multiple elements where social capital remained dominant. According to the social capital theory, the trustworthiness and norms of reciprocity are inseparable in a dense social network [52] and our study also found the interaction between the elements of social

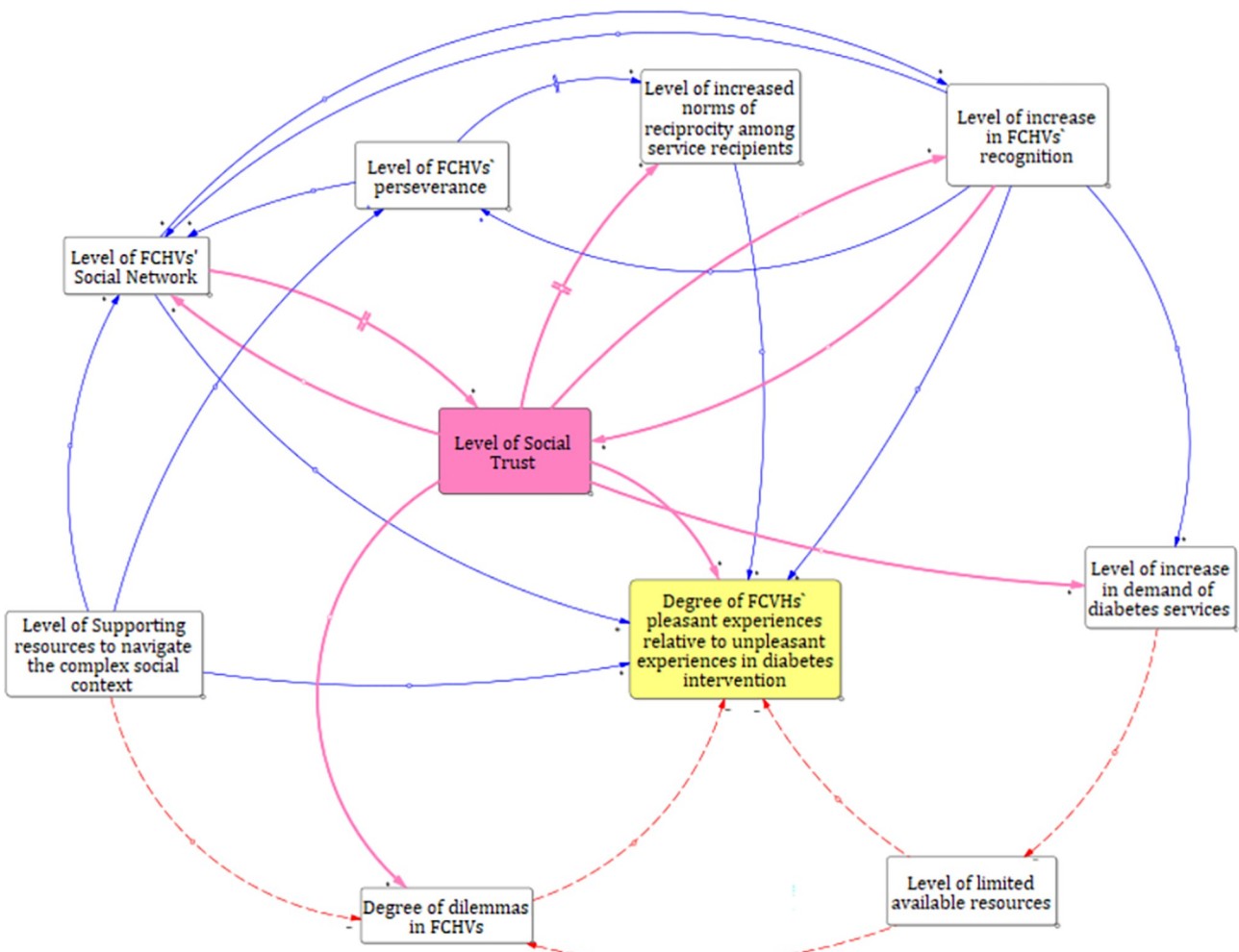

**Fig 2. Systems map of the interconnections and feedback loops influencing the degree of FCHVs' pleasant and unpleasant experiences in the diabetes intervention.** The solid blue lines show positive polarity, meaning, a variable is impacted in the same direction of change by another variable. For example, when social trust increases, recognition also increases. The dashed red lines show negative polarity, meaning, a variable is impacted in the opposite direction of change by another variable. For example, when the dilemma among FCHVs increases, their pleasant experience decreases. Parallel blue lines in the arrow show delayed effect, meaning, it takes some time to show the effect of a variable in another variable. For example, the level of reciprocity was shown quite late by the service recipients after a continuous dedication of FCHVs.

capital from the beginning of the T2D intervention, where the power dynamics [47] further added the complexity. However, the level of trust became an important element interconnecting with almost all elements shown in Fig 2.

Moreover, the systems maps (Figs 1 and 2) show that FCHV had to invest a huge amount of time and effort to deal with social phenomena rather than the technical aspects in diabetes intervention. We have discussed our findings mainly using the concept of social capital by Robert Putnam [52] including Bourdieu and Coleman [46, 47] to understand the holistic experiences of FCHVs in the phenomena. Further analysis of findings using systems thinking principles [48] showed the complex interconnections of social capital, recognition received by FCHVs for their work, resources available for the T2D, FCHVs'dilemmas and support systems impacting their un/pleasant experiences during their involvement in diabetes intervention (Fig 2).

## Social capital embedded with power dynamics (Bourdieu 1986)

**a) Gender.** Although FCHVs experienced mistrust from both the men and women at the beginning, mistrust from the men outweighed the mistrust from the women. Such mistrust from male members could be partly explained by their fresh relations or the male-dominating culture in a country like Nepal with a patriarchal society [53, 54]. Male skepticism on FCHVs' qualifications and capabilities to perform the basic T2D screening services fueled the hardships in expanding their social network, reinforced the community mistrust, and delayed the fostering norms of reciprocity among service recipients (Fig 1). The gender-based power dynamics was indicated by several phenomena, for example, FCHVs' experiences with some of the intimidating behavior from men led to mental stress, fear of the possibility of men's mis-behavior while walking alone long distances. Furthermore, our study findings aligned with a study among female health volunteers in India where men take the health decisions for their wives and halt the females service recipients to receive services from FCHVs [5]. In addition, the frequent use of the "he" pronoun in the interviews signals the possible symbolic violence resulting from power struggles and conflict that are not recognized. Such power struggles could get even harsher; for instance, female health volunteers in India have faced severe sexual harassment because of their mobility and public profile [55]. Thus, gender-based power dynamics demanded continuous effort of FCHVs to create, sustain and strengthen the trust in the community. To explain the intensity of gender-based power dynamics in the context of FCHVs' involvement in diabetes intervention is beyond the capacity of our study; however, further research is needed to explain these dynamics from both gender and psychological perspectives.

**b) Socio-economic status.** The existing power struggle in a caste-based society also led FCHVs to face accusations of excluding lower caste people in diabetes intervention (Fig 1). Such accusations from Dalits can be partially obvious as systematic reviews from Nepal and India also showed the Dalits had less opportunity for health service access because of their caste status and poverty [56]. Previous studies [14, 18] showed that the lower the educational status of community people, higher the chance of facing difficulties for CHWs to gain trust in regards to the health education they are providing. As opposed to this evidence, our study shows that middle-class families with comparatively high levels of education were predominantly skeptical of the match between FCHVs' quality services and their qualifications (see S1 Table).

Bourdieu explained the act of reciprocity also depends on economic capital [47]. The lack of reciprocity among the service recipients in the form of unwillingness to wait for FCHVs in the morning for screening could result from wage-based workers prioritizing their end meets over their health needs. Irrespective of the presence or absence of social trust, a study in Swe-den, revealed that people were obliged to act reciprocally as responsible members due to the fear of social exclusion [57]. Unlike Sweden's study, in our study, the fear of exclusion among wage-based workers may not have been prioritized over making ends meet, signaling the sig-nificance of economic status. Moreover, these wage-based workers may have felt that they were exempted from practicing the norms of reciprocity since there was no social obligation for them to participate in the diabetes intervention, for instance, fear of losing the bond with FCHVs. Despite the variance in the Swedish and Nepalese contexts, their findings have added some meaningful perspectives to our study in terms of how elements of social capital interplayed.

**c) Interconnectedness of elements of social capital (Putnam 2004) and beyond.** Fig 2 depicts that the level of social trust was increased with increase in the social network and vice-versa. Putnam explicitly emphasized that a social network can be a robust asset for both

individuals and communities, and its values have the capability to influence even bystanders [58]. This concept helps explain how FCHVs' existing (Mothers' group) and newly built social networks (bonding social capital) became an asset in spreading the information on their newly assigned tasks and how that circle sensitized the bystanders by increasing recognition towards FCHVs' work (Fig 2).

Likewise, social networks of community people contributed to the bystanders accessing information [44, 59] about the types of diabetes services provided by FCHVs. Expanding social networks by FCHVs alone, however, did not suffice the trust build-up process between FCHVs and the community people. The service recipients did confirmation tests at health centers in parallel to verify their blood sugar levels tested by FCHVs, which further yielded trust and increased recognition of their work. The systematic review among CHWs revealed that trust is the central issue and can only be achieved after deliberate action [14]. Our study showed that supporting resources (Fig 2): better bonding, bridging, and linking social capital helped FCHVs expand their social network, thus, gain the community's trust, which ultimately contributed to generating reciprocal feelings among many service recipients. A cross-sectional study in Lao People's Democratic Republic also showed better performance of village health volunteers when they have a better social capital [60]. Besides, FCHVs' dedication and multiple efforts to save people's lives may have positively influenced the trust build-up between community people and them.

Community people started accepting and acknowledging FCHVs' timeless efforts which was evident from their cooperative behaviors and the increased demand for diabetes screening, in the later phase (Figs 1 and 2). Moreover, active community involvement could improve FCHVs' performance and motivate them to continue working as volunteers [60]. Similar findings are supported by a review article showing the association of CHWs' motivation, retention, and accountability, with the level of support they receive, community embeddedness, and acceptance of their work in the community [11]. Thus, the meaningful engagement of community people in any global health interventions is now an important agenda to discuss in global health discourse.

## Level of supporting resources for FCHVs

FCHVs'supporting resources involved monetary incentives as well as social capital inclusive of bonding, bridging, and linking social capital. Various monetary and non-monetary resources supported FCHVs' perseverance in their work despite the multiple challenges. As described by Putnam, the norms of reciprocity [52] can be sensed from their motivation to serve the community as responsible community members and happiness gained from saving people's lives, and their strong belief in "Service as Religion,". Such motivation was sustained and amplified by the prestige and identity (level of increase in recognition, Fig 2) they received from the community people, which they sensed as their transformation from "ordinary FCHVs" to "professional FCHVs" in the community, further motivating them to persistently provide diabetes services [5] (Fig 2). India's female health volunteers' motivations were also reinforced by the respect, recognition, and prestige they received in the community [5]. Various other studies also show that the CHWs' work satisfaction sustains their motivation [5, 14, 17]. In addition, other studies also show the staff motivation, community support system, and social prestige as facilitators for CHW's work [1].

Another support system was the bonding social capital. FCHVs'family members, especially their husbands, were the strong support systems for FCHVs' motivation and retention in diabetes intervention as supported by study based on female CHWs in India and Bangladesh [5, 13, 17]. Likewise, linking social capital was also an important support system. As illustrated by

the timely logistical and technical support and supervision provided by NGO. Similar findings are shown by previous studies such as a systematic review [11], and the studies conducted among FCHVs in Nepal [28, 61]. Our findings reinforce the findings of Gyawali et al. that FCHVs are capable of delivering home-based basic diabetes services provided with appropriate training and having a strong support system [27]. However, several studies showed the pragmatic factors as the barriers to FCHVs in other national programs. For instance, limited trained human resources in national health systems [62], inadequate logistics [1, 62], workload, inadequate training, and remuneration, and poor support from health systems [1].

Some FCHVs considered the monetary incentives that they've received in T2D intervention as a form of recognition of their hardships, challenging some studies that have categorized financial incentives as an obvious extrinsic factor [61, 63, 64]. Nevertheless, some also felt that it covered basic operational costs such as communication and transportation expenses, which might have added motivations, as shown by other studies [11, 13, 65]. However, monetary incentives contradicted the principles of volunteerism as community people harshly accused FCHVs of working for money, especially at the program's start when the incentive amount was not transparent. Also, in Ethiopia, Kenya, Mozambique, and Malawi, volunteerism was found as a key element of positive relationships with community people [66]. If volunteerism is discussed with right-based perspectives to have a basic wage for CHWs [64, 67] because of the high demands expected from CHWs by stakeholders and community people in terms of time, flexibility, and workload [11, 14, 67], community-based health programs demand further ethical and right based discussion and research.

## FCHVs in dilemma

Interestingly, amidst the T2D intervention, FCHVs were also captured in a deep dilemma (Figs 1 and 2). FCHVs gaining the trust of community people generated new expectations among the community people, which required FCHVs to make some hard choices. They struggled to fulfill the community peoples' high demand to provide diabetes-related services due to program protocol, thus, the available resources were also limited. Hence, they felt it difficult to maintain the trust, reputation, and social relationships with their relatives, neighbors, and community, ultimately risking the depletion of the community's social capital [43]. Their relationship with family members and neighbors was at stake; thus, they were compelled to invest extra time and effort to satisfy the increased demand for diabetes screening and provide false hope to have patience and wait for their turn. Some FCHVs used their personal resources to calm the situations caused by the demand spike. For instance, FCHVs bought extra kits from medical shops (usually from their son's medical shops), extra batteries, and sought help from their relatives to use their private vehicles (usually motorbikes) to reach the venues for diabetes screening, including to accommodate service recipients to reach the health facilities. Similar findings were apparent from a in South Africa where CHWs had to constantly grapple for their recognition from "becoming" in the initial phase to "maintaining" in the later phase [14, 68]. In addition, FCHVs tried to satisfy the community people by measuring their blood pressure as they were also trained and equipped with a "blood pressure meter". The downside of this scenario would be the negative influence on their motivation and retention, affecting the program's efficacy. On the other hand, utilizing the "blood pressure meter" served as the opportunity to sensitize people about hypertension.

## Limitations of the study

Variations in the interview format could have led to some technical biases as some face-to-face interviews were switched to an online format due to the restrictions applied amid COVID-19.

Analyzing our findings based on the social capital theory provided valuable perspectives, yet Putnam himself has warned that social capital is insufficient to explain all phenomena [69]. However, we have tried to overcome this challenge by inductive coding and used the systems perspectives to find other dominating systems and connections in FCHVs'experiences (Figs 1 and 2).

## Conclusions

FCHVs' experiences in the T2D intervention were influenced by the interplay of the elements of social capital; social trust being the prominent intensified by the power dynamics related to gender and socio-economic status. Most of the challenges in T2D intervention were beyond the technical and logistic aspects; the challenges were raised from the social aspects. The challenges were expanding their new social network, building trust among the community people, and receiving the community support. The social capital, technical and financial resources played a significant role for FCHVs'perseverance to navigate the complex social context. It is apparent from our study that navigating the social challenges itself is not recognized as a part of the FCHVs'job. The social dimensions and FCHVs' experiences interacted in a dynamic way making feedback loops while influencing each other reflecting the complexity of social aspects that they had to navigate. From this study, we conclude that the social dimensions play an enormous role in delivering the community-based health intervention, which is usually not considered during the project cycle.

FCHVs' own monetary and non-monetary motivations were substantial factors to keep them determined towards their roles and responsibilities in T2D intervention. Thus, the role of social capital is noteworthy, yet their personal motivations are also embedded in the social capital. Future research could explore the interplay of elements of social capital and the success of CHWs' health interventions, including the state's role in those interventions. Trained FCHVs on the basic role of diabetes intervention could provide diabetes services at the community level. However, there is a need to recognize their additional roles in navigating the complex social structure that demanded their enormous time and efforts to gain the trust of the community people. Before implementing such programs, mental, emotional, and social stress needs to be considered, which may negatively affect their retention and sustain their motivation in the long run.

## Supporting information

**S1 Text. Topic guide for focus group discussion with FCHV (English version).**
(DOC)

**S2 Text. Topic guide for in-depth interview with FCHV (English version).**
(DOC)

**S1 Table. Study participants' demographic profile.**
(DOC)

## Acknowledgments

Special thanks to the study participants for their valuable contributions in the interviews. We thank the field researchers (Ms. Namrata Baral and Ms. Yamuna Kshetri) for assisting us in data collection. We acknowledge the World Diabetes Foundation (WDF) staffs: Ms. Marianne Kjærtinge Faarbæk, Ms. Elsa Morandat, and Ms. Mette Skar for providing technical

information about their project and organizing the knowledge-sharing session after the completion of the study.

## Author Contributions

**Conceptualization:** Usha Dahal, Rekha Lama Tamang, Dinesh Neupane.

**Data curation:** Usha Dahal, Rekha Lama Tamang, Sweta Koirala Adhikari, Pabitra Babu Soti.

**Formal analysis:** Usha Dahal, Rekha Lama Tamang.

**Funding acquisition:** Usha Dahal, Rekha Lama Tamang.

**Investigation:** Usha Dahal, Rekha Lama Tamang, Sweta Koirala Adhikari.

**Methodology:** Usha Dahal, Rekha Lama Tamang, Tania Aase Dræbel, Dinesh Neupane, Bishal Gyawali.

**Project administration:** Pabitra Babu Soti, Bishal Gyawali.

**Resources:** Usha Dahal, Rekha Lama Tamang, Dinesh Neupane, Sweta Koirala Adhikari, Pabitra Babu Soti, Bishal Gyawali.

**Software:** Usha Dahal, Rekha Lama Tamang.

**Supervision:** Tania Aase Dræbel, Bishal Gyawali.

**Validation:** Usha Dahal, Rekha Lama Tamang, Tania Aase Dræbel, Sweta Koirala Adhikari, Bishal Gyawali.

**Visualization:** Usha Dahal, Rekha Lama Tamang, Tania Aase Dræbel, Dinesh Neupane.

**Writing – original draft:** Usha Dahal, Rekha Lama Tamang.

**Writing – review & editing:** Usha Dahal, Rekha Lama Tamang, Tania Aase Dræbel, Dinesh Neupane, Sweta Koirala Adhikari, Pabitra Babu Soti, Bishal Gyawali.

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
